# Impact of the COVID-19 pandemic on emergency outpatient consultations and admissions of non-COVID-19 patients (ECCO) —A cross-sectional study

**Nina Hangartner** [1] * , **Stefania Di Gangi** [2] , **Christoph Elbl** [1], **Oliver Senn** [2], **Fadri Bisatz** [1], **Thomas Fehr** [1]

**1** Department of Internal Medicine, Cantonal Hospital Graubünden, Chur, Switzerland, **2** Institute of Primary Care Medicine, University Hospital Zurich, Zurich, Switzerland

☯ These authors contributed equally to this work.
* nina.hangartner@ksgr.ch

## Abstract

During the first year of the COVID-19 pandemic, healthcare facilities worldwide struggled to adequately care for the increasing number of COVID-19 patients while maintaining quality of care for all other patients. The aim of this study was to investigate the displacement and underuse of non-COVID-19 patient care in a medical department of a tertiary hospital in Switzerland. In this retrospective cross-sectional study, internal medicine admissions from 2017 to 2020, emergency outpatient visits from 2019 to 2020 and COVID-19 admissions in 2020 were analyzed and compared using a regression model. Internal medicine admissions were also stratified by diagnosis. A questionnaire was used to assess the pandemic experience of local general practitioners, referring hospitals, and nursing homes. The total number of admissions decreased during the 1st and 2nd waves of the pandemic but increased between the two waves. Elective admissions decreased in 2020 compared to pre-pandemic years: they represented 25% of total admissions in 2020 versus 30% of the total admissions during 2017–2019, p <0.001. Admissions for emergency reasons increased: 71% in 2020 versus 65% in 2017–2019, p < 0.001. Emergency outpatient consultations decreased in 2020 compared to 2019, 62.77 (14.70), mean (SD), weekly visits in 2020 versus 74.13 (13.98) in 2019, p<0.001. Most general practitioners and heads of referring hospitals also reported a decrease in consultations, especially during the 1st wave of the pandemic. Mental illnesses, anxiety or burn-out were perceived in both patients and staff in general practices and nursing homes. In conclusion, the COVID-19 pandemic negatively affected the care of non-COVID-19 patients, particularly those with chronic illnesses. A shift of health care resources from non-COVID patients to COVID patients was observed. These findings could help institutions better manage such a situation in the future.

**Data Availability Statement:** All relevant data are within the paper and its Supporting information files.

**Funding:** The author(s) received no specific funding for this work.

**Competing interests:** The authors have declared that no competing interests exist.

# Introduction

## Background

In December 2019, a new coronavirus called SARS-CoV-2 (severe acute respiratory syndrome coronavirus 2) first appeared at a local fish and wildlife market in Wuhan City, China. Within three months, the virus had spread to countries around the world, prompting the World Health Organization (WHO) to declare the SARS-CoV-2 outbreak a global pandemic in March 2020 [1].

The first cases in Switzerland were recorded in February 2020. Government measures were taken to contain the further spread of the virus, resulting in a national lockdown from March 17 to April 26. Public and private gatherings of more than five people were prohibited, schools and most businesses were closed, and home office was recommended. Thereafter, there was a temporary nationwide decrease in COVID-19 cases until October 2020, when a 2nd wave of infection took place [2].

In early 2020, the healthcare system in the Swiss region (canton) of Graubünden, where our study took place, needed to take action to provide the necessary resources to care for COVID-19 patients in the upcoming pandemic. Elective medical care was substantially reduced, special wards and outpatient clinics for the treatment of COVID-19 patients as well as testing centers were established and staffed, and an additional intensive care unit was set up. This new organization, faced with unforeseen requirements, posed a major challenge to the health system and its staff.

In everyday clinical life we observed that, despite the new coronavirus disease, the total number of hospitalizations and consultations in the Internal Medicine department decreased, and fewer patients visited the emergency room. This is consistent with the perception and statistics of other healthcare institutions worldwide during the pandemic. In the Department of Veterans Affairs, the largest healthcare system in the United States, reported a 41% decline in admissions due to emergencies in the first 16 weeks of 2020 compared to the same time period in 2019 [3]. In Germany, the number of patients admitted to the emergency room declined by approximately 30% from February to April 2020 [4]. In Italy, hospitalizations for myocardial infarction halved in March 2020 [5], and in the United States, cardiac catheterization laboratory activations for STEMI (ST-Segment Elevation Myocardial Infarction) decreased by 38% [6]. Neurological and cancer-related outpatient emergencies and admissions also decreased in France and Turkey [7–9].

## Objectives

In this study, changes in admissions and outpatient emergency consultations in the Department of Internal Medicine at Cantonal Hospital Graubünden, Switzerland, during the COVID-19 pandemic (from January 2020 to December 2020) were evaluated and compared with those of the three previous years. The goal was to determine the impact on medical care for non-COVID-19 patients. Surveys for general practitioners, nursing homes, and referring hospitals were used to describe the observed changes and shifts and to understand the experiences of different healthcare providers with the pandemic.

# Materials and methods

## Study design

In this retrospective cross-sectional study, the impact of the COVID-19 pandemic on admissions to the general internal medicine wards and outpatient consultations in the emergency room of the medical Department in the Cantonal Hospital of Graubünden, was

investigated in comparison to the three previous years (2017–2019). General medicine at the Cantonal Hospital of Graubünden includes 99 acute care beds and 14 beds in each palliative care and geriatrics. Weekly patient admissions to the medical department were recorded as elective or emergency admissions and classified according to the primary admitting diagnosis.

## Data collection and data quality

**Admissions and emergency outpatient consultations.** Hospitalization data from January 1, 2017, to December 31, 2020, were extracted from the discharge database of the Cantonal Hospital of Graubünden. The medical records included: case number, patient's date of birth, gender, principal diagnosis, department, whether it was an elective admission or an emergency, admission date, and discharge date. Information about emergency department consultations in the years 2019 and 2020 was obtained from the emergency department database itself and included: case number, date of birth, gender, department, and date of visit. For 2017 and 2018, the diagnosis codes of emergency department outpatient consultations were not yet systematically registered because no such policies existed. These years could therefore not be considered for the analysis.

**COVID-19 admissions.** Hospitalizations of COVID-19 patients in the pandemic ward, in the intensive care unit and patients admitted for other reasons but with a positive COVID-19 at admission or during hospitalization were recorded in an additional EXCEL database by our chief physician. The medical records included only the date of hospitalization and were recorded without disclosing patient identity.

**Surveys.** Three different questionnaires were created for general practitioners (GPs), heads of referring hospitals, and nursing homes administrators, using an Internet-based program (SurveyMonkey). The questionnaires consisted of eight to twenty German questions specific for each group.

A combination of question types was used, including mainly closed questions, but also questions that allowed a free-text answer. All questions that were translated into English, are listed in the Supplementary S1 File.

The surveys were conducted in February and March 2021. Respondents participated voluntarily and all responses were anonymized.

An access link to the survey was sent by e-mail to all 68 general practitioners (GPs) near the Cantonal Hospital of Graubünden. A total of 40 completed questionnaires were received, corresponding to a response rate of 59%. Of the 11 heads of referring hospitals, all completed the questionnaire, corresponding to a response rate of 100%. A total of 11 nursing homes were invited, of which 8 completed the questionnaire, resulting in a response rate of 73%.

**Study size.** The sample size consisted of 5311 total admissions on average for each year in 2017–2020 and 3628 outpatient consultations for 2019 and 2020.

## Data description and outcomes

The primary outcome was the number of weekly admissions (apw), analyzed by department wards, reason of admission, and patient characteristics. The department wards were: General Medicine, Geriatrics, Palliative Care, and Pandemic, the latter for 2020 only. The type of admission was emergency or elective. Principal diagnoses were categorized using International Classification of Diseases–10th (ICD-10) codes [10]. In subgroup analyses, the following categories were considered: diseases of the circulatory system (I00-I99), diseases of the respiratory system (J00-J99) and malignant neoplasms (C00-C99). In addition, ICD codes I20-I24 (acute ischemic heart disease), I25 (chronic ischemic heart disease), I63 (cerebrovascular disease),

J00-J06 (acute upper respiratory infections), J09-J18 (influenza and pneumonia), J20-J22 (other acute lower respiratory infections), J40-J47 (chronic lower respiratory diseases) and COVID-19 admissions were analyzed separately.

Secondary outcome was the number of weekly emergency department outpatient consultations (cpw).

Primary and secondary outcomes were compared between two groups: pandemic year (2020) and pre-pandemic years (2017–2019).

Patient gender and age were considered in the descriptive analysis.

The 1st wave of the pandemic was defined from calendar week 10 to calendar week 24. The 2nd wave of the pandemic occurred from calendar week 39 to the end of the year.

The following data from GPs survey were analyzed: total workload % during the lockdown, between the 1st and 2nd waves and during the 2nd wave of the pandemic, compared to the workload before the lockdown; workload tasks (i.e. consultations in practice, home visits, tele-consultations . . .) as % of the total workload during the lockdown, compared to the situation before the lockdown; perceptions about patients and quality of care during the pandemic, defined as yes/no questions and free text answers. The following information was collected from the nursing homes: the number of patients cared for, the form of medical care organization (home physician, mixed model. . .), perceptions about patients and quality of care during the pandemic, defined as yes/no questions. Data were collected from referring hospitals on changes in admissions (increase, reduction, no changes) and perceptions of quality of care (yes/no questions and open questions).

A minimal dataset to reproduce the main findings was provided in S1 Dataset.

## Statistical methods

In descriptive tables for admissions and survey, continuous variables were reported as mean (SD: standard deviation) or median [IQR: interquartile range]; categorical variables as N, number of cases (%). The Chi-Square test was used for group comparison (pandemic versus pre-pandemic) of categorical/binary variables and the t-test or non-parametric Kruskal-Wallis-Test for continuous variables.

To compare average weekly admissions and consultations in the pandemic and pre-pandemic groups, separately for the 1st and 2nd wave periods, we used robust t-test comparisons of means reporting pairwise absolute differences with 95% bootstrap (500 replicates) confidence interval and p-values. The results were shown graphically by a dumbbell plot or connected dot plot, which is an alternative to a grouped bar graph, to better show the changes between two time points.

The weekly number of admissions and consultations, in the pandemic and pre-pandemic groups, was estimated using a negative binomial regression model, in line with other studies [11–16]. To adjust for the two pandemic waves, in spring and fall/winter, we used a three-basis spline model with an interaction term with the pandemic and pre-pandemic group. Trends of admissions were analyzed overall and separately for type of admission and subgroups of diagnoses. Results of the regression models were presented graphically. Risk ratio (RR) with 95% CI (confidence intervals) for the predictors and other details of the models were provided in S1 Table. For the analysis of COVID-19 recordings, results were presented graphically as a smoothed curve.

A significance level of $P \leq 0.05$ applied to all analyses. All statistical analyses were performed using the statistical package R version 4.1.0, from R Foundation for Statistical Computing, Vienna, Austria, https://www.R-project.org/.

The survey results were transferred from the online platform "Survey Monkey" into an Excel spreadsheet (Microsoft) and then imported in R. Qualitative data analysis from surveys, i.e., free text answers, was also performed.

### Ethics

In accordance with the cantonal ethics committee Zurich, an approval was not needed for this retrospective, non-interventional study (BASEC-Nr. Req-2020-00759).

## Results

### Patient and admission characteristics

Table 1 represented a descriptive analysis of the weekly admissions by period: pre-pandemic (years 2017–2019) and pandemic (year 2020). Overall, no significant difference was found in the average weekly number of admissions between pandemic and pre-pandemic period, 99.08 (20.43) versus 100.59 (14.81), p = 0.560.

**Table 1. Admissions and emergency outpatient consultations by period, type of admissions and patient characteristics.** P are the p-values of Chi-Square test (categorical/binary variables) and t-test for mean values (continuous variables). Significant values, p≤0.05, were denoted in bold.

| | Admission period | | |
|---|---|---|---|
| | Pre-pandemic Years 2017–2019 | Pandemic year 2020 | p |
| **Hospital Admissions** | | | |
| n | 15,994 | 5251 | |
| Number of weekly admissions | | | |
| mean (SD) | 100.59 (14.81) | 99.08 (20.43) | 0.560 |
| **Patient characteristics** | | | |
| ICD[a] code I, n (%) | 5415 (33.9) | 1625 (30.9) | **<0.001** |
| I20-I24, n (%) | 1348 (8.4) | 448 (8.5) | 0.837 |
| I25, n (%) | 715 (4.5) | 176 (3.4) | **0.001** |
| I63, n (%) | 567 (3.5) | 211 (4.0) | 0.123 |
| ICD code J, n (%) | 1327 (8.3) | 545 (10.4) | **<0.001** |
| ICD code C, n (%) | 2151 (13.4) | 725 (13.8) | **0.526** |
| Other ICD codes, n (%) | 7101 (44.4) | 2356 (44.9) | 0.563 |
| Male gender, n (%) | 9028 (56.4) | 2997 (57.1) | 0.435 |
| Not survived, n (%) | 1037 (6.5) | 371 (7.1) | 0.150 |
| Age (years), mean (SD) | 67.22 (16.31) | 67.97 (16.25) | **0.004** |
| **Admission characteristics** | | | |
| Geriatrics, n (%) | 749 (4.7) | 264 (5.0) | 0.327 |
| General medicine, n (%) | 14,347 (89.7) | 4415 (84.1) | **<0.001** |
| Palliative care, n (%) | 898 (5.6) | 328 (6.2) | 0.095 |
| Emergency, n (%) | 10,385 (64.9) | 3724 (70.9) | **<0.001** |
| Elective, n (%) | 4777 (29.9) | 1298 (24.7) | **<0.001** |
| Pandemic, n (%) | - | 244 (4.6) | - |
| **Emergency outpatient consultations[b]** | | | |
| n | 3929 | 3327 | |
| Number of weekly visits | | | |
| mean (SD) | 74.13 (13.98) | 62.77 (14.70) | **<0.001** |

[a] International Classification of Diseases–10th.

[b] For emergency outpatient consultations, only data for year 2019 was considered as pre-pandemic period.

Over time, there was significant decrease in the proportion of admissions to the General Medicine ward to all admissions: 4415 (84%) in 2020 versus 14,347 (90%) in the pre-pandemic period, p<0.001, while admissions to the wards of Geriatrics and Palliative care did not change.

On average, patients admitted in the pandemic year 2020 were significantly older compared to the pre-pandemic period: 67.97 (16.25) versus 67.22 (16.31) years of age, p = 0.004.

## Total admissions: Trends during the pandemic

The average number of weekly admissions (apw) in 2020 decreased significantly by about 9% compared to the pre-pandemic years during the first wave (absolute difference, rounded to the nearest unit, -9 apw, p = 0.04, Fig 1), but not during the second wave. Seasonal trends were also observed. In the 1st and 2nd waves, weekly cases decreased, whereas in the summer months, weekly cases increased. (Fig 2A).

## Emergency and elective admissions

The proportion of emergency admissions to total admissions was significantly higher in 2020 than in other pre-pandemic years: 3724 (71%) versus 10,385 (65%), p < 0.001, Table 1. On the other hand, the proportion of admissions for elective reasons to total admissions was significantly lower: 1298 (25%) versus 4777 (30%), p <0.001.

Weekly emergency admissions, increased significantly in 2020 between the 1st and before the 2nd wave, Fig 2B, but not during the pandemic waves compared with the pre-pandemic years (Fig 1).

In contrast, weekly elective admissions in 2020 decreased in the 1st and 2nd waves, Fig 2C, with an absolute difference, between the pandemic and pre-pandemic averages, of -9 apw,

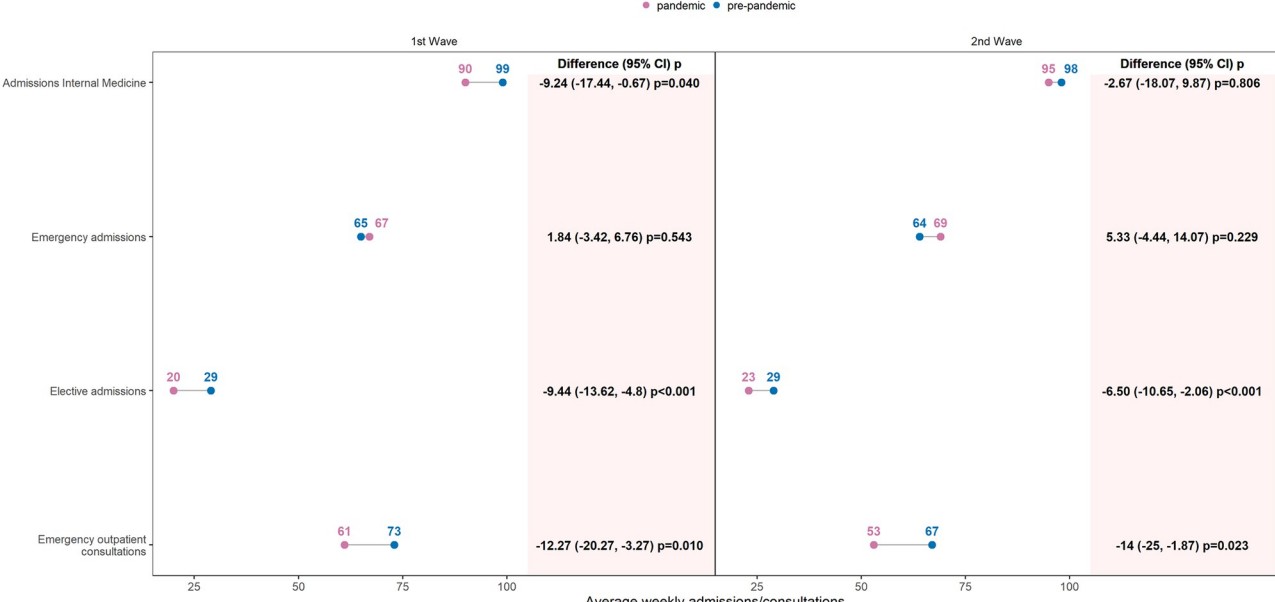

**Fig 1. Yearly changes in pandemic year (2020) weekly admissions and emergency outpatient consultations, compared to pre-pandemic years (2017–2019).** Connected dots represented the average number of weekly admissions and emergency outpatient consultations in pre-pandemic versus pandemic period during the two pandemic waves. Differences between the two means and 95% bootstrap Confidence Interval (CI) with p-values (robust t-test) were indicated in columns. 1st Wave: calendar weeks (10–24); 2nd Wave: calendar weeks (39–53).

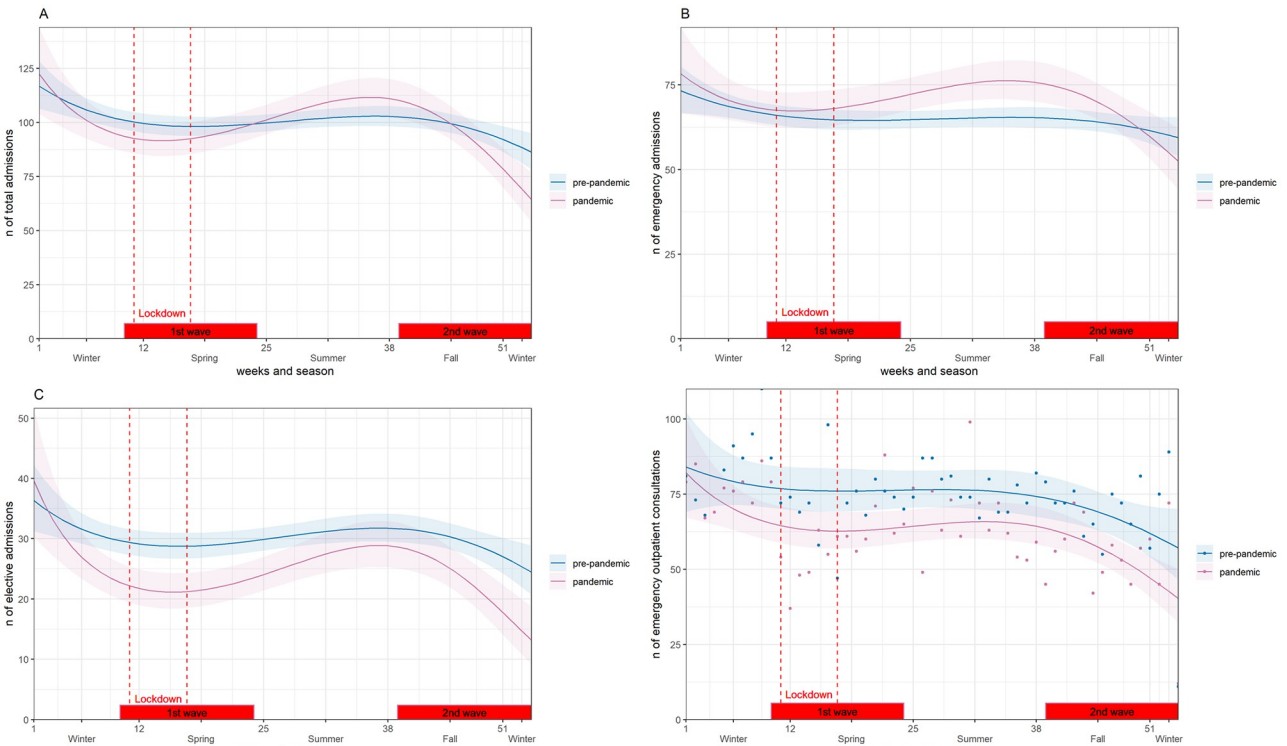

**Fig 2. Weekly changes in admissions and emergency consultations from 2017–2019 (pre-pandemic period) and 2020 (pandemic period).** Negative binomial regression model of number of admissions by calendar week, year and type of admission. (A) Admissions Internal Medicine. (B) Emergency admissions. (C) Elective admissions. (D) Emergency outpatient consultations. 1st Wave: calendar weeks (10–24); 2nd Wave: calendar weeks (39–53). Lockdown: March 17—April 26.

p<0.001, decrease of 32%, in the 1st wave and of -6 apw, p = 0.001, decrease of 22%, in the 2nd wave, (Fig 1).

## Outpatient emergency consultations

In 2020, mean weekly outpatient emergency consultations in Internal Medicine were significantly lower than emergency outpatient consultations in 2019 (pre-pandemic period): 62.77 (14.70) versus 74.13 (13.98), p<0.001, Table 1.

During the 1st wave of the COVID-19 pandemic, weekly cases were 16% lower compared to 2019 (absolute difference -12 cpw, p = 0.010) and of 21% lower in the 2nd wave (-14 cpw, p = 0.023), Fig 1. Looking at the trend over weeks an increase in cases was observed in 2020 in the first month following the lockdown (Fig 2D).

## Admissions by diagnosis groups

**ICD-code I (diseases of the circulatory system).** There were significantly fewer admissions for patients with ICD code I as a proportion of total admissions in 2020 compared to pre-pandemic years: 1625 (31%) versus 5415 (34%), p <0.001, Table 1. During the 1st wave, in 2020, the weekly admissions decreased significantly by 26% compared to the pre-pandemic period, absolute difference -9 apw, p<0.001 and during the 2nd wave by 18%, 6 apw, p = 0.027, Fig 3. Seasonal differences within 2020 were also observed. In the summer of 2020, weekly admissions increased to reach the level of the previous years, Fig 4A.

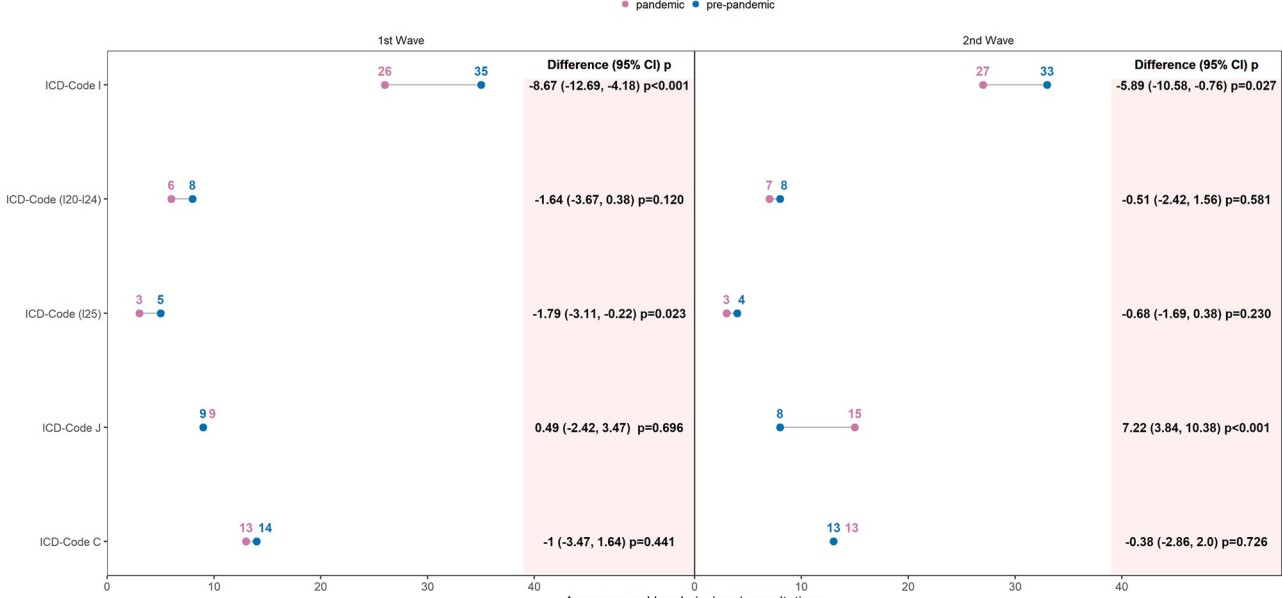

**Fig 3. Yearly changes in pandemic year, 2020, weekly admissions by diagnosis, compared to pre-pandemic years (2017–2019).** Connected dots represented the average number of weekly admissions by International Classification of Diseases–10th (ICD) code group in each period, pandemic and pre-pandemic, during the two pandemic waves. The differences between the two averages and 95% bootstrap Confidence Interval (CI) with p-values (robust t-test) were reported in columns. 1st Wave: calendar weeks (10–24); 2nd Wave: calendar weeks (39–53).

Additionally, we analyzed the acute ischemic heart diseases (ICD I20-24) and the chronic ischemic heart diseases (ICD I25) separately and found a decreasing trend in admissions during the 1st and 2nd waves and an increase in admissions between the 1st and 2nd waves in 2020 in both subgroups (Fig 4B). Comparing averages between pandemic and pre-pandemic groups, the difference was significant for ICD I25 during the 1st wave, -2 apw, p = 0.023, 40% reduction, Fig 3.

**ICD-code J (respiratory system).** The proportion of ICD-Code J admissions to total admissions was significantly higher in 2020 than in pre-pandemic years: i.e. 545 (10%) versus 405 (7%), p<0.001, Table 1. In the 1st wave of the COVID-19 pandemic in 2020, no significant change, p = 0.687, in weekly admissions was observed compared to the same time period in pre-pandemic years, Fig 3. In the second half of the year, weekly admissions increased and remained significantly higher than in previous years (Fig 4C). During the 2nd wave, average weekly admissions were 87% higher in 2020 compared to pre-pandemic years with an absolute difference of 7 apw, p<0.001, Fig 3. Similar trends were found for ICD codes of lower respiratory infection (S1 Fig).

**ICD-code C (malignant neoplasm).** There was no significant difference in weekly admissions for the ICD-Code C in 2020 compared to the pre-pandemic years, during the 1st and 2nd waves, Fig 3. There was a decreasing trend in the 1st wave, an increase in the summer in 2020, and a decrease again in the 2nd wave (Fig 4D).

## COVID-19 admissions

The first admissions for COVID-19 to the Cantonal Hospital Graubünden occurred in March 2020, that is at the beginning of the 1st wave of the pandemic. During the summer months,

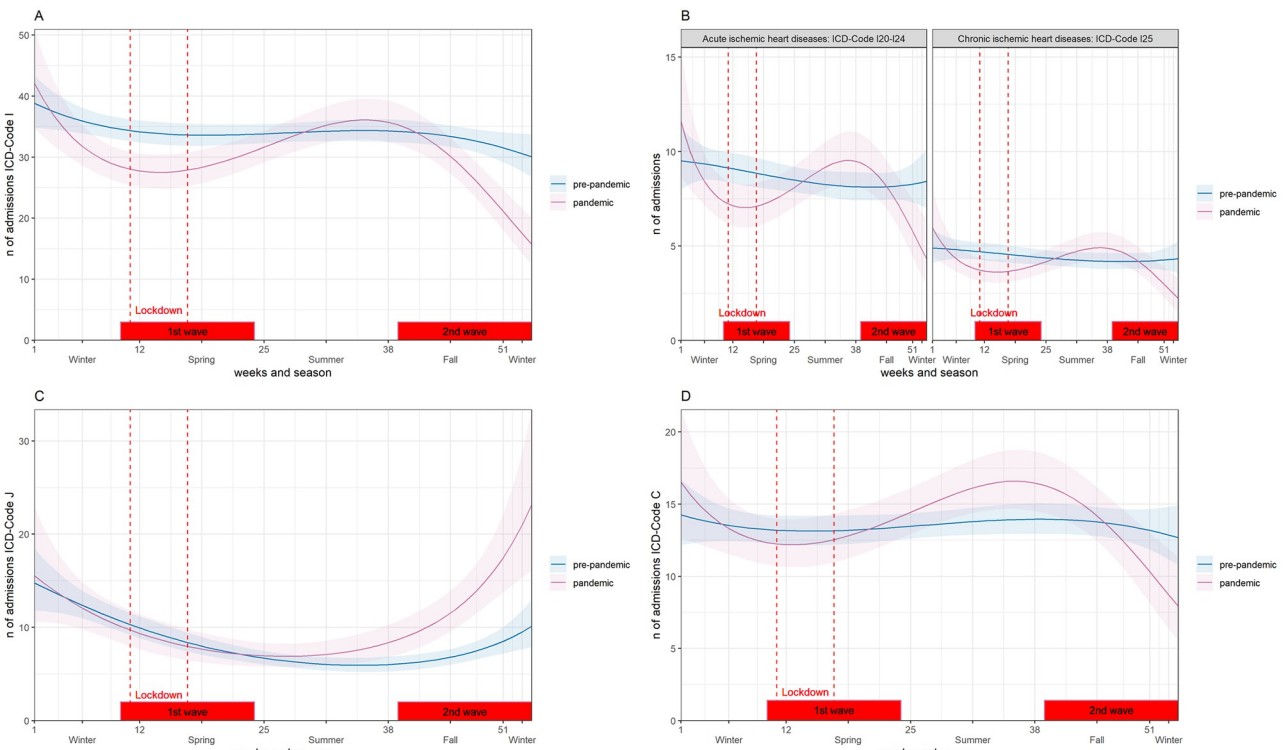

**Fig 4. Weekly changes in admissions by diagnosis from 2017–2019 (pre-pandemic period) and 2020 (pandemic period).** Negative binomial regression model of the number of admissions by International Classification of Diseases–10[th] (ICD) code. (A) Circulatory system ICD-Code I. (B) Ischemic heart diseases ICD-Codes I20-24 and I25. (C) Respiratory system ICD-Code J. (D) Neoplasm ICD-Code C. 1st Wave: calendar weeks (10–24); 2nd Wave: calendar weeks (39–53). Lockdown: March 17—April 26.

weekly admissions decreased until beginning of fall, when they increased again, defining the 2nd wave of the pandemic (Fig 5).

## Surveys

**General practitioners (GPs).**   General practitioners workload decreased during the lockdown: with a reference workload of 100 before the lockdown, half of the GPs reported a workload of 70% during the lockdown, median [IQR]: 70% [55%, 80%], p<0.001 (Table 2). The reduction of workload was related to consultations in the practice. Instead, the workload for home visits, tele-consultations, homecare, and organizational aspects doubled compared to the situation before the lockdown.

About half of the GPs 22 (55%) reported that patients with chronic diseases received inadequate medical care during the pandemic and were in poorer health due to limited consultations. All doctors had patients who cancelled appointments, and 55% of the practitioners cancelled non-urgent appointments themselves. Two-thirds 26 (65%) of the GPs reported that patients refused a medically indicated referral to a hospital. Due to the pandemic, the majority of GPs 33 (82%) saw more patients with mental illnesses, and in 27 (67%) of the medical practices wrote advance directives more frequently (Table 2).

From the qualitative analysis of free text answers, the following criticisms or considerations from GPs emerged: "I consider centralized care in the Cantonal Hospital for moderately severe

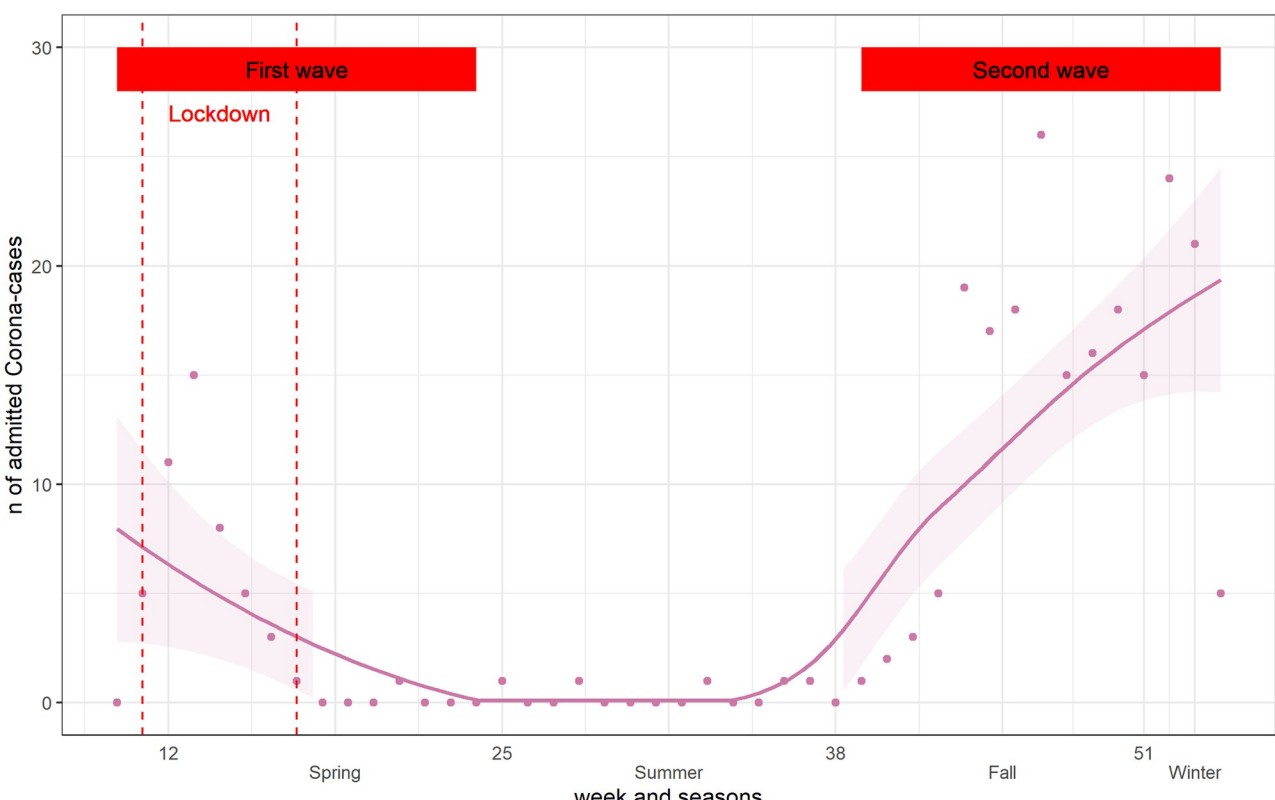

**Fig 5. COVID-19 admissions by calendar week.** Points represented observed values and line was the smoothed curve. 1st Wave: calendar weeks (10–24); 2nd Wave: calendar weeks (39–53). Lockdown: March 17—April 26.

COVID disorders (as in the second wave) to be important in order to provide high quality COVID care centrally. In this way, elective procedures will have to be postponed less frequently at peripheral hospitals". "New requirements and regulations every day create more confusion than they help". "Patients tended to be discharged from the hospital even more quickly, leading to difficult situations in the outpatient setting". "Some weaknesses in the pandemic strategy are political rather than medical". "Social isolation, especially in nursing homes, is very problematic." "The federal government's recommendation to discourage visits to the GP was devastating. Hospitals should not have closed outpatient clinics and certain elective procedures." "The waiver of non-urgent consultations was not necessary." "However, the Cantonal Hospital was well organized".

**Heads of referring hospitals.** During the 1st wave of the pandemic, the majority 10 (91%) of heads of referring hospitals reported a decrease in admissions, and all hospitals recorded a reduction in emergency outpatient consultations. During the 2nd wave, only 3 (27%) hospitals reported a decrease in admissions, while 4 (36%) reported no change and 4 (36%) reported an increase in admissions. Three (27%) hospitals reported an increase and three a decrease in emergency outpatient consultations, and consultations remained constant in 5 (45%) hospitals (Table 2).

*Nursing homes.* On average, a nursing home cared for about 91 patients. Two (25%) reported cases of patients refusing referral to a hospital. Most nursing homes 7 (87%) did not reduce the number of physician visits for their residents. About two-third of nursing homes

**Table 2. Descriptive table of survey results.** Lockdown: March 17—April 26. 1st Wave: calendar weeks (10–24); 2nd Wave: calendar weeks (39–53).

| | General Practitioners (GPs) n = 40 | | | | |
|---|---|---|---|---|---|
| | **Before lockdown** | **Lockdown** | **Between 1st -2nd waves** | **2nd wave** | **p** |
| **Total Workload (%), median [IQR]** | 100 | 70 [55 80] | 100 [90 100] | 100 [90, 100] | <0.001 |
| **Tasks as % oft the total Workload, median [IQR]** | | | | | |
| Consultation hours in the practice | 80 [67.50, 80] | 50 [40, 60] | | | <0.001 |
| Home visits (excluding home care) | 5 [5, 10] | 5 [5, 10] | | | 0.136 |
| Telephone consultations | 5 [5, 10] | 20 [10, 20] | | | <0.001 |
| Organizational issues | 5 [5, 5] | 10 [10, 20] | | | <0.001 |
| Home care | 5 [5, 10] | 10 [5, 10] | | | 0.747 |
| Other | 7.50 [5, 10] | 10 [5, 12.50] | | | 0.452 |
| **Question / Perception:** | | | During the pandemic | | |
| Shortage of care for chronically ill patients | Yes | N(%) | 22 (55.0) | | |
| Patients in worse health condition | Yes | N(%) | 20 (50.0) | | |
| Patients cancelling regular appointments | Yes | N(%) | 40 (100.0) | | |
| GP cancelling regular appointments | Yes | N(%) | 22 (55.0) | | |
| Patients refusing hospitalization | Yes | N(%) | 26 (65.0) | | |
| GP reducing hospitalizations | Yes | N(%) | 7 (17.5) | | |
| Patients with more mental health problems | Yes | N(%) | 33 (82.5) | | |
| More patient advance directives | Yes | N(%) | 27 (67.5) | | |
| | Nursing homes n = 8 | | | | |
| | | | During the pandemic | | |
| **Number of patients, mean (SD)** | 91.12 (55.35) | | | | |
| **Medical Care Organisation, N(%)** | | | | | |
| Home physician | 3 (37.5) | | | | |
| Former family physicians of the patients | 1 (12.5) | | | | |
| Mixed model | 4 (50.0) | | | | |
| **Question / Perception:** | | | | | |
| Patients refusing hospitalization | Yes | N(%) | 2 (25.0) | | |
| Less medical consultations | Yes | N(%) | 1 (12.5) | | |
| More patient advance directives | Yes | N(%) | 4 (50.0) | | |
| Patients with more mental health problems | Yes | N(%) | 6 (75.0) | | |
| Staff suffering isolation, anxiety, burn out | Yes | N(%) | 7 (87.5) | | |
| | Referring hospital n = 11 | | | | |
| | | | **1st wave** | **2nd wave** | **p** |
| **Hospital admissions, N(%)** | | | | | 0.008 |
| No change | | | 1 (9.1) | 4 (36.4) | |
| Reduction | | | 10 (90.9) | 3 (27.3) | |
| Increase | | | 0 (0) | 4 (36.4) | |
| **Emergency consultations, N(%)** | | | | | 0.002 |
| No change | | | 0 (0) | 5 (45.5) | |
| Reduction | | | 11 (100) | 3 (27.3) | |
| Increase | | | 0 (0) | 3 (27.3) | |

reported that the preferred course of action in the event of a medical problem was discussed more frequently with residents, and advance directives were written more frequently in half of the nursing homes. Residents in 6 (75%) of the nursing homes and staff in 7 (87%) were more likely to suffer from mental illness (Table 2).

## Discussion

### Admissions and trend overall

Overall, weekly admissions to the General Medicine department decreased in 2020 compared to the pre-pandemic period. Seasonal differences were observed in 2020: cases decreased during the 1st and 2nd waves of the COVID-19 pandemic, while weekly cases increased during the summer month. Such a decline followed by an increase in cases after the 1st wave has been described in two previous studies from the United States, which found that hospitalizations declined in March and April 2020 and a rebound effect was observed thereafter [17, 18]. Whether the increase of patients between the two waves in our hospital was a rebound effect or delayed visits of chronically ill patients we could not exactly know. However, the results of the following GP survey suggested that it might have been a combination of both.

### Elective vs emergency admissions

During the first two waves of the pandemic, elective admissions declined, likely due to hospitals postponing elective admissions to save resources. Specifically, a 32% decrease was observed during the 1st wave compared to weekly elective admissions in the pre-pandemic years. We presumed that the observed increase in elective admissions, at the time COVID-19 cases temporarily declined between the two waves, could be explained by a rebound of previously delayed outpatient consultations and procedures. In fact, it appeared that patients' health was deteriorating and they needed to be admitted more frequently.

The survey evidenced that the majority of primary care patients and a minority of home nursing patients refused hospitalization. Instead, the majority of general practitioners did not suggest avoiding hospital care. Accordingly, in a survey including 5,412 adults, conducted in the United States [19], 41% of the respondent adults reported having delayed or avoided medical care. However, emergency department admissions as a percentage of total admissions increased in 2020, in contrast to emergency department outpatient consultations, which actually decreased in 2020.

### COVID-19 impact on other diseases

For admissions due to cardiovascular problems, the decrease was relevant in the 1st and 2nd waves: i.e. around 26% in the 1st wave compared to weekly admissions in the previous year. In part, this change could be due to deferred elective coronary angiography, as admissions increased again after the 1st wave. A decreasing trend in admissions for acute cardiovascular diagnosis was found during the 1st and 2nd waves. This result corresponds to the finding of a systematic review [20]. A similar seasonal trend was observed in admissions for malignant neoplasm. Accordingly, a French study [21] described a 20% decrease in breast cancer diagnosis during lockdown, followed by a 48% increase after the lockdown, with delay leading to enlarged tumors and worse diagnosis.

In addition, we found that the overall number of respiratory disease in 2020 increased significantly, particularly during the 2nd wave of the pandemic, relative to the increased number of COVID-19 cases admitted and due to the fact, that respiratory diseases are always more frequent in winter month [22]. Worldwide one could observe that the second wave of the pandemic showed higher numbers of COVID-19 patients. There is evidence of a decrease in respiratory diseases due to restrictions imposed to contain the pandemic: for example, in Taiwan [23] influenza, enterovirus, and all-cause pneumonia decreased overall when masks, social distancing, or hand disinfection were introduced; in Korea [24], seasonal influenza activity in 2020 decreased substantially compared to previous years. A study of respiratory illness in

children in Alaska [25] also found a decrease in hospitalizations for non-COVID respiratory illness compared to previous years. Our data could only show similar trends in the first wave of the pandemic but showed an increase of overall respiratory diseases in the second wave. ICD-Codes J09-J18, which include also COVID-19 cases increased to nearly 20 cases per week. This corresponded to the number of COVID-19 cases, shown in Fig 5.

## Outpatient emergency consultations

Emergency outpatient consultations in 2020 decreased significantly compared to the previous year, similar to what has been found in several studies around the world. In our study, emergency outpatient consultations decreased by 16% and 21% in the first and second waves, respectively, compared to the weekly admissions in the previous year. Neurology emergency department visits in Saudi Arabia decreased by 24% in the first month of the pandemic compared to pre-pandemic, hospital emergency department consultations in Germany decreased by 32% in March and April 2020 compared to February 2020, and in the United States decreased by 40% from March 15 to May 23 2020, compared to the preceding 10-week period. [4, 7, 26]. In the survey we conducted, referring hospitals and general practitioners also reported a decrease in their workload, especially during the 1st wave of the pandemic.

## Impact on medical care services and non-COVID patients

General practitioners reported a decrease in workload during the 1st wave, while workload returned to pre-pandemic levels during the second wave. Consistent with these findings, another study in Switzerland found evidence of a decline in the number of consultations in general practice [27]. In our survey, GPs reported that not only did patients canceling appointments, but some GPs themselves were reducing appointments. Accordingly, general practitioners in New Zealand [28] experienced a decline in consultations, particularly at the beginning of the pandemic, with negative consequences for certain chronic conditions and screening, and a resulting backlog of work.

General practitioners and nursing homes reported that patients and staff suffered more frequently from mental illnesses. A study in France [29] showed an increase in mental health problems among GPs during the first lockdown in France. Accordingly, a cross-sectional survey study in 8 European countries from April to June 2020 [30] revealed the impact of the pandemic on mental health, with higher rates of depression, anxiety, and stress among medical and non-medical professionals. Based on these findings, remedial strategies have been proposed to address these problems [31].

In addition, patients appear to be more concerned with end-of-life decisions, with GPs and nursing homes reporting more frequent discussions about preferred course of action in the event of medical problems and about living wills.

All general practitioners reported patients canceling regular appointments, and half of the practitioners felt that chronically ill patients suffered from inadequate medical care and that their health deteriorated because of delayed consultations. Early in the pandemic, Lisa Rosenbaum pointed out these effects in her article [32], and they have been discussed in several other studies [19, 28, 33, 34]. Consistent with these findings, GPs in our study cited fear of COVID infection in medical settings and avoidance of burdening the health care system as the two main reasons patients postponed or canceled appointments despite medical problems.

From the GPs free-text answers, criticism emerged regarding the shift of resources, the postponement of procedures and government directives which discouraged patients to visit the GP. In our study, critics represented 12.5% of the participating GP. In a German study [35] four types of opinion patterns among GP, regarding COVID-19 restrictions, could be

identified: Corona 'Sceptics', 'Hardliners', 'Balancers', and 'Anxious'. Corona Sceptics, which could be compared to critics in our study, represented 8% of the participating GP. Unfortunately, only a minority of GPs answered the free-text question. We could suppose that the other participants fully agreed, accepted, did not complain or have suggestions about the pandemic strategy during the lockdown. According to one GP the centralized care for COVID-19 patients, as organized in the Cantonal Hospital of Graubünden during the second wave, helped to provide high quality of care and peripheral hospitals to maintain capacity for elective procedures and treatment of non-COVID patients.

### Reallocation of medical care resources

As mentioned above the overall admissions decreased during the first and second wave of the pandemic. Consequently, staff could be relocated to a specialized ward for COVID-19 patients. The same effect could be observed in in the emergency room where neither we had staff shortages.

### Strengths and limitations

The greatest strength of the study is the integration of quantitative analysis of a central hospital data with descriptive and qualitative analysis of healthcare providers' perception of the pandemic. In fact our study connected together results of changes in hospital admissions to changes in workload and quality of care in general practices and referring hospitals.

By the different nature and source of data, the two analyses were separate but since GPs and heads of referring hospitals were located in the same area of the central hospital, their respective results are meant to be complementary as different points of view of the same issue. Moreover, in our data, all diagnoses have a reliably assigned ICD-Code.

On the other hand, our study has also some limitations. First and foremost, this is a study at a single center A definitive statement can therefore only be made about the local conditions and the layers of care around the Cantonal Hospital of Graubünden. Within the entire canton Graubünden, however, our findings are supported by the results of the questionnaires sent to the referring physicians and the chief physicians of the regional referring hospitals. There is also some evidence of similar observations in the international literature, suggesting that the trends described have not only taken place in Graubünden [7, 17, 19, 20, 28, 32, 36].

Another important limitation is not having COVID-19 testing information in every patient was admitted. Therefore, we could not determine whether admissions for non-COVID-related respiratory illnesses were in fact due to COVID-19. This strategy was later changed during the $2^{nd}$ to $5^{th}$ wave, where all admitted patients were systematically tested. Lastly, patients were not surveyed directly. In this study, inferences about patients' decisions were drawn from the results of the questionnaires sent to referring physicians and institutions.

### Implications for research and/or practice

Though existing research shows similar patterns and findings, this is the first study to focus on changes in admissions and outpatient emergency consultations, in Switzerland, during the COVID-19 pandemic from January 2020 to December 2020, compared with those of the three previous years, which is a large interval with respect to the one considered in most of the studies. In addition, the study gives an overview of the impact on non-COVID patient care, experienced by other healthcare providers. This may influence decisions for health care resource allocation in a future pandemic.

## Conclusions

Our analysis found that the COVID-19 pandemic negatively impacted the care of non-COVID-19 patients, particularly those with chronic conditions. Thus, there was a shift in health care resources from non-COVID patients to manage the 1st and 2nd waves of the pandemic. This unplanned re-allocation of resources is concerning and raises questions, particularly because pandemic may last longer than expected and because a significant proportion is unvaccinated against COVID-19 at both the national and international level. For further pandemics a centralized high-quality care of infected patients is important, while peripheric hospitals and general practitioners should be able to continue regular appointments and the care of all other patients. Governments communication should focus on encouraging patients with chronic illnesses to attend their regular medical visits.

Health care providers and politicians should be aware of this resource shift and bring the issue of unequal distribution of health care resources into the public debate.

## Supporting information

**S1 File. List of all survey questions.**
(PDF)

**S1 Dataset. Minimal data set to reproduce the main analysis.**
(XLSX)

**S1 Table. Details of the negative binomial models showed in Figures.** BS1, BS2, BS3 denoted the basis splines, used in the non-linear model, depending on calendar week. Risk ratio (RR) 95% CI (confidence intervals) were reported for each predictor.
(DOCX)

**S1 Fig. Weekly changes in admissions for respiratory system from 2017–2019 (pre-pandemic period) and 2020 (pandemic period).** Negative binomial regression model of the number of admissions by International Classification of Diseases–10[th] (ICD) code. Subgroup analysis. J00-J06 (acute upper respiratory infections), J09-J18 (influenza and pneumonia), J20-J22 (other acute lower respiratory infections), J40-J47 (chronic lower respiratory diseases) 1st Wave: calendar weeks (10–24); 2nd Wave: calendar weeks (39–53). Lockdown: March 17—April 26.
(TIF)

## Acknowledgments

We would like to thank Dr. med Marina Jamnicki, physician official of the Canton Graubünden, and Mrs. Stephanie Casanova, from civil registry office Plessur for providing data for our study. Further thanks go to all the general practitioners, heads of referring hospitals and nursing homes for responding to our survey.

## Author Contributions

**Conceptualization:** Nina Hangartner, Stefania Di Gangi, Christoph Elbl, Thomas Fehr.

**Data curation:** Nina Hangartner, Stefania Di Gangi, Fadri Bisatz.

**Formal analysis:** Nina Hangartner, Stefania Di Gangi.

**Investigation:** Nina Hangartner, Fadri Bisatz.

**Methodology:** Nina Hangartner, Stefania Di Gangi, Christoph Elbl, Oliver Senn, Thomas Fehr.

**Project administration:** Nina Hangartner, Thomas Fehr.

**Resources:** Oliver Senn, Fadri Bisatz.

**Software:** Stefania Di Gangi.

**Supervision:** Christoph Elbl, Oliver Senn, Thomas Fehr.

**Visualization:** Stefania Di Gangi.

**Writing – original draft:** Nina Hangartner, Stefania Di Gangi, Christoph Elbl.

**Writing – review & editing:** Stefania Di Gangi, Christoph Elbl, Oliver Senn, Thomas Fehr.

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
