## [Decision Letter · Decision Letter 0]

21 Mar 2022

PONE-D-22-06720Impact of the COVID-19 pandemic on emergency outpatient consultations and admissions of non-COVID-19 patients (ECCO) – a cross-sectional studyPLOS ONE

Dear Dr. Hangartner,

Thank you for submitting your manuscript to PLOS ONE. After careful consideration, we feel that it has merit but does not fully meet PLOS ONE’s publication criteria as it currently stands. Therefore, we invite you to submit a revised version of the manuscript that addresses the points raised during the review process.

We look forward to receiving your revised manuscript.

Kind regards,

Yong-Hong Kuo

Academic Editor

PLOS ONE

Journal Requirements:

a) Did participants provide their written or verbal informed consent to participate in this study?

Additional Editor Comments:

Overall, the referees find this work interesting and important. I agree with the referees and believe this is a timely and significant topic. However, there are some concerns regarding the details of the study, which should be clarified before the consideration for publication.

The referees have provided constructive comments. Please incorporate them in the revision.

Reviewers' comments:

Reviewer's Responses to Questions

**Comments to the Author**

1. Is the manuscript technically sound, and do the data support the conclusions?

Reviewer #1: Yes

Reviewer #2: Yes

Reviewer #3: Partly

2. Has the statistical analysis been performed appropriately and rigorously? 

Reviewer #1: Yes

Reviewer #2: Yes

Reviewer #3: No

3. Have the authors made all data underlying the findings in their manuscript fully available?

Reviewer #1: Yes

Reviewer #2: Yes

Reviewer #3: Yes

4. Is the manuscript presented in an intelligible fashion and written in standard English?

Reviewer #1: Yes

Reviewer #2: Yes

Reviewer #3: Yes

5. Review Comments to the Author

Reviewer #1: Dear Editor,

Only few problems need to be clarified in this manuscript.Some previous studies showed delayed or rebounding effects of visiting hospitals among chronic illnesses because of shortages of health care resources and the risk of infection. How about the reallocation of medical care resources during the COVID-19 pandemic in your hospital? Were there shortages of health care resources for the non-COVID-19 disease? The total number of admission increased between the two waves of the pandemic, and was it the rebounding effect or delayed effect among patients with chronic illness?

Reviewer #2: The topic is meaningful and insightful. The related statistic analysis is well-rounded. The data visualizations and tables are clearly presented. The framework is logical and well-organized. Some additional comments are as follows:

(1) Line 136-137. The response rate and the number of questionnaires are not quantitatively high. Will more surveys make the final results more reliable?

(2) Line 176-178. The Poisson regression model is used. Some literature review or data analysis could be added, in order to explain the reason for using the Poisson regression model. (Such that, the fact that previous papers had conducted similar modelling, or the verification that dataset follows Poisson distribution, etc.)

(3) Line 323-325. Not very rigorous. Here, the reduction is considered for "later in the pandemic and during the 2nd wave". However, in the 2nd wave, there is no apparent workload reduction. Also, regarding the workload for home visits, teleconsultations, homecare, and organizational aspects, not all of these quantities are doubled. All of these quantities are not shown in the 2nd wave, based on table 2.

(4) In terms of the Results/Surveys and Discussion sections, the statistic or survey findings and corresponding analysis are separated into these two sections, respectively. This is fine, but the analysis in the Discussion section has not covered all the patterns in the Results/Surveys section. For example, what is the consequence of free-text answers listed in Line 333-344?

(5) The authors mentioned the strength of this study in Line 457-458. Quantitative analysis refers to the statistics and qualitative analysis are from surveys. However, the integration of these two parts is not strong.

Currently, these two parts are quite separated. Please enhance the descriptions of the integration.

(6) As shown in the literature reviews, the author mentioned some existing research have revealed similar patterns(Line 476-477) in other countries. In this way, could the author please emphasize the innovative points of this study(such as the cross-sectional study, or the phenomenons especially in Switzerland)? Otherwise, this paper is more likely to be a report, rather than a research article.

(7) This paper has presented good statistics and surveys but has not provided profound support of resource allocation, which is one of the keywords.

Reviewer #3: Reviewer’s comments:

The authors performed a cross-sectional study to evaluate the impact of the COVID-19 pandemic. The results are interesting. However, several shortcomings were found. These questions are listed as follows:

1. L. 75-77. Since the decrease in patient number is a common phenomenon, the meaning of these sentences may confuse. Please rewrite and delete the word “interestingly”.

2. L.116-117. Concerning the data credibility, please explain why the data of 2017 and 2018 are unreliable.

3. L.119-122. The recording of COVID-19 patients’ information was different from that of other patients. Please explain the reasons and the differences. In addition, the authors mentioned that the medical records of the COVID-19 patients only included the date of hospitalization. How did these patients' ICD codes record? Did the recording method cause misclassification bias?

4. P.13. Table 1. The readers expect to see the differences caused by the pandemic, not the comparisons among 2019 and prior years (e.g. ICD code C). The statistical method should focus on the variation caused by the pandemic.

5. L.396-398. The subject of this discussion was chronic diseases, but the reference cited was about acute diseases. Please correct.

6. The authors had collected ICD codes of acute cardiovascular diseases. However, no such discussion was found. The authors are recommended to add acute cardiovascular diseases in the discussion section.

7. L.405-407. The authors mentioned that the number of respiratory diseases differed from other reports. Was it due to the recording method of the COVID-19 patients? (question 3)?

8. The authors are advised to select the patients with ICD codes of lower respiratory infection for subgroup analysis.

9. The author mentioned that differences between the first and second waves existed (e.g. L.404). More analysis and discussion about these differences should be provided.

10. Statistical analysis is the strength of this study, while the qualitative part gave little information. The authors are advised to strengthen the statistical part (e.g. more diseases and other variables) and deepen the discussion (e.g. the cause of the phenomenon).

11. In the discussion section, the authors should propose more concrete and feasible ways to mitigate the impact of a further wave.

6. PLOS authors have the option to publish the peer review history of their article (what does this mean?). If published, this will include your full peer review and any attached files.

Reviewer #1: No

Reviewer #2: No

Reviewer #3: No

---

## [Author Response · Author response to Decision Letter 0]

29 Apr 2022

Point by point reply

Reviewers’ comments

Reviewer #1:

Only few problems need to be clarified in this manuscript. Some previous studies showed delayed or rebounding effects of visiting hospitals among chronic illnesses because of shortages of health care resources and the risk of infection. How about the reallocation of medical care resources during the COVID-19 pandemic in your hospital? Were there shortages of health care resources for the non-COVID-19 disease? The total number of admission increased between the two waves of the pandemic, and was it the rebounding effect or delayed effect among patients with chronic illness?

We thank the reviewer for the comment. We have addressed these important questions in the discussion.

- In our hospital we had the possibility to create a separate emergency „department“ for patients with symptoms of COVID-19, and we could also open a separate ward for hospitalized COVID-19 patients. These measures could better relocate resources among patients and help to reduce the risk of an in-hospital infection.

- Luckily, we never had shortages of material or room for patients. Staff shortages existed primarily in the intensive care unit.

- Whether the increase of patients between the two waves was a rebound effect or delayed visits of chronically ill patients we could not exactly know. However, the results of the GP survey suggested that it might have been a combination of both.

Reviewer #2

The topic is meaningful and insightful. The related statistical analysis is well-rounded. The data visualizations and tables are clearly presented. The framework is logical and well- organized.

We thank the reviewer for this encouraging feedback.

Some additional comments are as follows:

 (1) Line 136-137. The response rate and the number of questionnaires are not quantitatively high. Will more surveys make the final results more reliable?

We sent the questionnaires to all general practitioners, chiefs of hospitals and nursing homes surrounding our hospital. We also sent a reminder in case of no answer. The response rates to the survey (59% for GPs, 100% for medical heads for regional hospitals, and 73% for nursing homes) was quite acceptable. However, the reviewer is right, that more surveys might increase the reliability, but our hospital is located within a rather small canton with a regular population of only around 200,000 inhabitants, but fluctuations of up to 100’000 depending on touristic season. Therefore, we could not recruit and invite more doctors to

participate.

 (2) Line 176-178. The Poisson regression model is used. Some literature review or data analysis could be added, in order to explain the reason for using the Poisson regression model. (Such that, the fact that previous papers had conducted similar modelling, or the verification that dataset follows Poisson distribution, etc.)

We thank the reviewer for this comment. Many measures of healthcare use are event counts, for example, number of hospital admissions. Poisson regression is commonly used for count data. In particular, it was used for hospital admissions in other studies which we have added in the reference list. This is why we used it. However, Poisson regression is based on given assumptions. In case of overdispersion, it is better to use alternative like negative binomial or quasi-poisson. Therefore, we have revised the models using negative binomial instead of Poisson model. However, the main conclusions of our study are still valid, though the confidence intervals of the curves are different.

The reviewer is right. We apologize for not having spotted this inaccuracy. We have removed “Later in the pandemic and during the 2nd wave, only a slight reduction was reported” since, as reported in Table 2, the median workload is 100, between the two waves and during the second wave. Regarding the workload for home visits, teleconsultations, homecare, and organizational aspects, we have this information only before the lockdown and during the lockdown (see question 2, for general practitioners, in S1 File.pdf)

We thank the reviewer for this comment. We have added in the discussion some considerations about free-text answers.

 (3) Line 323-325. Not very rigorous. Here, the reduction is considered for "later in the pandemic and during the 2nd wave". However, in the 2nd wave, there is no apparent workload reduction. Also, regarding the workload for home visits, teleconsultations, homecare, and organizational aspects, not all these quantities are doubled. All these quantities are not shown in the 2nd wave, based on table 2.

 (4) In terms of the Results/Surveys and Discussion sections, the statistic or survey findings and corresponding analysis are separated into these two sections, respectively. This is fine, but the analysis in the Discussion section has not covered all the patterns in the Results/Surveys section. For example, what is the consequence of free-text answers listed in Line 333-344?

 (5) The authors mentioned the strength of this study in Line 457-458. Quantitative analysis refers to the statistics and qualitative analysis are from surveys. However, the integration of these two parts is not strong.

Currently, these two parts are quite separated. Please enhance the descriptions of the integration.

We thank the reviewer for the suggestion. We have tried to better highlight the integration, though by nature and source of data, the two analyses are separate.

 (6) As shown in the literature reviews, the author mentioned some existing research have revealed similar patterns (Line 476-477) in other countries. In this way, could the author please emphasize the innovative points of this study (such as the cross-sectional study, or the phenomenons especially in Switzerland)? Otherwise, this paper is more likely to be a report, rather than a research article.

The reviewer is right. We have emphasized the innovative points in “Implications for research and/or practice” subsection.

(7) This paper has presented good statistics and surveys but has not provided profound support of resource allocation, which is one of the keywords.

The reviewer is right. Therefore, we have removed “resource allocation” from the keywords.

 Reviewer #3

The authors performed a cross-sectional study to evaluate the impact of the COVID-19 pandemic. The results are interesting. However, several shortcomings were found. These questions are listed as follows:

1. L. 75-77. Since the decrease in patient number is a common phenomenon, the meaning of these sentences may confuse. Please rewrite and delete the word “interestingly”.

We thank the reviewer for this comment. We have deleted the word “interestingly”.

2. L.116-117. Concerning the data credibility, please explain why the data of 2017 and 2018 are unreliable.

It was only in 2019 that a systematic registration of outpatients visit in the emergency department with categorization in “medical” and “surgical” outpatients was started. Therefore, any numbers before that time point are not reliable enough for comparison.

3. L.119-122. The recording of COVID-19 patients’ information was different from that of other patients. Please explain the reasons and the differences. In addition, the authors mentioned that the medical records of the COVID-19 patients only included the date of hospitalization. How did these patients' ICD codes record? Did the recording method cause misclassification bias?

The medical records of COVID-19 patients were identical to any other patient. However, since COVID-19 has no separate ICD code, we maintained a separate Excel list of the daily number of COVID-19 cases in the ICU, in the pandemic ward and in the emergency department.

 4. P.13. Table 1. The readers expect to see the differences caused by the pandemic, not the comparisons among 2019 and prior years (e.g. ICD code C). The statistical method should focus on the variation caused by the pandemic.

As already specified in the manuscript “Table 1 represented a descriptive analysis of the weekly admissions”. In the revised version, we have defined years 2017-2019 as the pre- pandemic period, to better identify the variations caused by the pandemic, The differences caused by the pandemic are illustrated in Figures 1 and 3. As explained in the methods, to compare average weekly admissions in 2020 with those in previous years, separately for the 1st and 2nd wave periods, the results were shown graphically by a dumb-bell plot or connected dot plot. For the weekly admissions, we used a negative binomial regression model as already used in other research we quoted.

We understand the reviewer point and we have added some details in the statistical method. We prefer to use the graphical and intuitive visualization instead of the more formal one, in order to be more comprehensible for a non-statistician reader. Anyway, the models were rigorously performed, and all assumptions tested, in line with the literature.

5. L.396-398. The subject of this discussion was chronic diseases, but the reference cited was about acute diseases. Please correct.

We thank the reviewer for this observation. We changed the title in “COVID-19 impact on other diseases” to discuss cardiovascular diseases and respiratory diseases that were not only chronic diseases but also acute (i.e. respiratory infections).

6. The authors had collected ICD codes of acute cardiovascular diseases. However, no such discussion was found. The authors are recommended to add acute cardiovascular diseases in the discussion section.

We thank the reviewer for this comment. Acute cardiovascular diseases were in fact already mentioned when citing the relative systematic review “According to our results.”. However, the reviewer is right, and we have better and explicitly discussed our results.

7. L.405-407. The authors mentioned that the number of respiratory diseases differed from other reports. Was it due to the recording method of the COVID-19 patients? (question 3)?

The number of respiratory diseases differed from other reports because the data of the overall number of respiratory diseases could also include COVID-19 cases. In fact, since we did not have COVID test information for all admissions, we could not know if a patient admitted for a respiratory disease may have COVID during hospital stay. We have clarified this point in the limitations.

8. The authors are advised to select the patients with ICD codes of lower respiratory infection for subgroup analysis.

We thank the reviewer for the advice. We have done as suggested and reported the additional analysis in S1 Fig.

9. The author mentioned that differences between the first and second waves existed (e.g. L.404). More analysis and discussion about these differences should be provided.

We thank the reviewer for the comment. We have deepened the discussion

10. Statistical analysis is the strength of this study, while the qualitative part gave little information. The authors are advised to strengthen the statistical part (e.g. more diseases and other variables) and deepen the discussion (e.g. the cause of the phenomenon).

We thank the reviewer for the comment. We have deepened the discussion and improved the statistical models. Anyway, we aimed to present our study in the way also a non-statistician could understand and therefore we preferred to not give the main role to the statistics or provide too many details of it in text (however provided as supplemental). As for the qualitative part, we think that it helps to better understand the observed changes in emergency consultations and admissions in the tertiary care hospital of the region by including GP perceptions on one side and experiences of the regional health care institutions on the other side.

11. In the discussion section, the authors should propose more concrete and feasible ways to mitigate the impact of a further wave.

We thank the reviewer for this comment. We have done as suggested.

---

## [Decision Letter · Decision Letter 1]

16 May 2022

PONE-D-22-06720R1Impact of the COVID-19 pandemic on emergency outpatient consultations and admissions of non-COVID-19 patients (ECCO) – a cross-sectional studyPLOS ONE

Dear Dr. Hangartner,

Thank you for submitting your manuscript to PLOS ONE. After careful consideration, we feel that it has merit but does not fully meet PLOS ONE’s publication criteria as it currently stands. Therefore, we invite you to submit a revised version of the manuscript that addresses the points raised during the review process.

We look forward to receiving your revised manuscript.

Kind regards,

Yong-Hong Kuo

Academic Editor

PLOS ONE

Journal Requirements:

Additional Editor Comments (if provided):

Overall, the referees are satisfied with the revision. Reviewer 3 has some suggestions for the authors to improve the work. I recommend Minor Revision.

Reviewers' comments:

Reviewer's Responses to Questions

**Comments to the Author**

1. If the authors have adequately addressed your comments raised in a previous round of review and you feel that this manuscript is now acceptable for publication, you may indicate that here to bypass the “Comments to the Author” section, enter your conflict of interest statement in the “Confidential to Editor” section, and submit your "Accept" recommendation.

Reviewer #1: All comments have been addressed

Reviewer #2: All comments have been addressed

Reviewer #3: All comments have been addressed

2. Is the manuscript technically sound, and do the data support the conclusions?

Reviewer #1: Yes

Reviewer #2: Yes

Reviewer #3: Yes

3. Has the statistical analysis been performed appropriately and rigorously? 

Reviewer #1: Yes

Reviewer #2: Yes

Reviewer #3: Yes

4. Have the authors made all data underlying the findings in their manuscript fully available?

Reviewer #1: Yes

Reviewer #2: Yes

Reviewer #3: Yes

5. Is the manuscript presented in an intelligible fashion and written in standard English?

Reviewer #1: Yes

Reviewer #2: Yes

Reviewer #3: Yes

6. Review Comments to the Author

Reviewer #1: Dear Editor,

The manuscript is meaningful and interesting. The authors had replied to all questions well.

Reviewer #2: (No Response)

Reviewer #3: Thank you for your revisions to the manuscript. I feel that you addressed almost all of my previous questions. However, I have a concern with this study as follows:

1.The authors mentioned that the number of respiratory diseases increased during the second wave of the pandemic. Because of no COVID-19 testing information, it could not determine whether it is due to the increase in COVID-19 cases. However, the authors mentioned in the limitation section that the COVID-19 test has been carried out since the second wave. Is it possible to know whether the COVID-19 cases caused the increase of respiratory diseases in the second wave? Please explain that more.

7. PLOS authors have the option to publish the peer review history of their article (what does this mean?). If published, this will include your full peer review and any attached files.

Reviewer #1: No

Reviewer #2: No

Reviewer #3: No

---

## [Author Response · Author response to Decision Letter 1]

20 May 2022

Point by point reply

Reviewers’ comments

Reviewer #3:

1.The authors mentioned that the number of respiratory diseases increased during the

second wave of the pandemic. Because of no COVID-19 testing information, it could not

determine whether it is due to the increase in COVID-19 cases. However, the authors

mentioned in the limitation section that the COVID-19 test has been carried out since the

second wave. Is it possible to know whether the COVID-19 cases caused the increase of

respiratory diseases in the second wave? Please explain that more.

We thank the reviewer for the comment.

We thank the Reviewer for this comment. We have clarified in the manuscript that our data showed an increase of ICD-Codes J09-J18, which include also COVID-19 cases shown in Figure 5. ICD-Code J12.8, generally used for viral Pneumonia, sometimes was also used for COVID. In addition, patients more severely ill with COVID, for example, with sepsis, again have a different code. Accordingly, an evaluation and direct comparison with the ICD codes could be biased. This is why, we collected and analyzed separately COVID cases. Therefore, we cannot determine how much the % of increase in all of respiratory diseases is due to COVID but we can speculate that this increase is due mainly to COVID cases, since COVID cases increase together with ICD-Codes J09-J18 which contain them. Moreover, looking at Figure 5 and S1 Fig there is evidence that the increase in ICD-Codes J09-J18 is mostly determined by the increase in COVID cases.

---

## [Decision Letter · Decision Letter 2]

27 May 2022

Impact of the COVID-19 pandemic on emergency outpatient consultations and admissions of non-COVID-19 patients (ECCO) – a cross-sectional study

PONE-D-22-06720R2

Dear Dr. Hangartner,

We’re pleased to inform you that your manuscript has been judged scientifically suitable for publication and will be formally accepted for publication once it meets all outstanding technical requirements.

Kind regards,

Yong-Hong Kuo

Academic Editor

PLOS ONE

Additional Editor Comments (optional):

Reviewers' comments:

Reviewer's Responses to Questions

**Comments to the Author**

1. If the authors have adequately addressed your comments raised in a previous round of review and you feel that this manuscript is now acceptable for publication, you may indicate that here to bypass the “Comments to the Author” section, enter your conflict of interest statement in the “Confidential to Editor” section, and submit your "Accept" recommendation.

Reviewer #3: All comments have been addressed

2. Is the manuscript technically sound, and do the data support the conclusions?

Reviewer #3: Yes

3. Has the statistical analysis been performed appropriately and rigorously? 

Reviewer #3: Yes

4. Have the authors made all data underlying the findings in their manuscript fully available?

Reviewer #3: Yes

5. Is the manuscript presented in an intelligible fashion and written in standard English?

Reviewer #3: Yes

6. Review Comments to the Author

Reviewer #3: The authors replied that COVID-19 cases in the second wave were recorded in an additional dataset without patients' identities (L.119-120). I believed the authors had already tried their best to answer my question.

7. PLOS authors have the option to publish the peer review history of their article (what does this mean?). If published, this will include your full peer review and any attached files.

Reviewer #3: No

---

## [Editor Report · Acceptance letter]

2 Jun 2022

PONE-D-22-06720R2 

Impact of the COVID-19 pandemic on emergency outpatient consultations and admissions of non-COVID-19 patients (ECCO) – a cross-sectional study 

Dear Dr. Hangartner:

I'm pleased to inform you that your manuscript has been deemed suitable for publication in PLOS ONE. Congratulations! Your manuscript is now with our production department. 

Kind regards, 

on behalf of

Dr. Yong-Hong Kuo 

Academic Editor

PLOS ONE